# A population-based study of tuberculosis incidence among rheumatic disease patients under anti-TNF treatment

Natália Sarzi Sartori[1]*, Paulo Picon[2], Afonso Papke[1], Jeruza Lavanholi Neyeloff[3], Rafael Mendonça da Silva Chakr[1,2]

**1** Department of Rheumatology, Hospital de Clínicas de Porto Alegre (HCPA), Universidade Federal do Rio Grande do Sul (UFRGS), Porto Alegre, Brazil, **2** Department of Internal Medicine, UFRGS, Porto Alegre, Brazil, **3** Planning and Evaluation Advisory Office, Hospital de Clínicas de Porto Alegre (HCPA), Porto Alegre, Brazil

\* nataliasartori2007@yahoo.com.br

**Data Availability Statement:** All relevant data are within the manuscript.

**Funding:** This work was funded by the Graduate Program in Medical Sciences of the Federal

## Abstract

### Introduction

Tuberculosis (TB) is an infectious disease caused by *Mycobacterium tuberculosis*. The advent of immunobiologic therapy with TNF inhibitors agents, has been associated with a significant increase in incident cases of tuberculosis in this population.

### Objective

To estimate the incidence of tuberculosis in patients receiving TNF inhibitors therapy for rheumatic diseases. As secondary objectives, we sought to evaluate mortality and the clinical impact of screening for latent tuberculosis infection.

### Methods

This retrospective study included patients with rheumatic diseases of Public Health System from the Brazilian state, a high TB incidence area, who received prescriptions of TNF inhibitors agents between 2006 and 2016.

### Results

A total of 5853 rheumatic disease patients were included. Patients were predominantly women (68.7%) aged 49.5 (± 14.7) years old. Forty-three cases of TB were found (2.86 cases per 1000 person-years; 18 times higher than in the general population). Adalimumab and certolizumab users presented a higher risk for TB development compared to etanercept users (RR: 3.11, 95%CI 1.16–8.35; 7.47, 95%CI 1.39–40.0, respectively). In a subgroup of patients, screening for latent tuberculosis infection was performed in 86% of patients, and 30.2% had a positive tuberculin skin test. Despite latent TB treatment, TB was diagnosed in 2 out of 74 (2.7%) patients. Overall, TB diagnosis did not increase mortality.

University of Rio Grande do Sul, through the
Research and Events Incentive Fund (FIPE) of the
Hospital de Clínicas de Porto, which provided
financial support for the publication of this study.
The funder had no role in study design, data
collection and analysis, publication decision, or
manuscript preparation.

**Competing interests:** The authors have declared
that no competing interests exist.

## Conclusion

In this population-based study of rheumatic disease patients from a high incident area, TNF inhibitor exposure was associated with an 18-time increased TB incidence. Adalimumab and certolizumab were associated with greater and earlier TB diagnosis compared to etanercept.

## Introduction

Tumor necrosis factor-alpha (TNF-α) is a cytokine involved in the pathogenesis of several systemic rheumatic diseases.[1,2] With the advent of TNF inhibitors therapy, treatment of these diseases has advanced markedly and clinical outcomes have improved, especially in patients refractory to conventional therapy.[3–7] The benefits of TNF inhibitors therapy have been well established in several studies that have demonstrated efficacy in controlling disease activity in rheumatoid arthritis (RA), ankylosing spondylitis (AS), psoriatic arthritis (PsA), and juvenile idiopathic arthritis (JIA).[8–10] Five TNF inhibitors agents are currently available for use in Brazil: infliximab (IFX), etanercept (ETN), adalimumab (ADA), golimumab (GOL), and certolizumab pegol (CZP).

TNF-α is known to play a role in the control of infectious diseases, particularly those caused by intracellular microorganisms such as *Mycobacterium tuberculosis*.[11] Its role is particularly important in organizing the activation and maintenance of granuloma.[12,13] Accordingly, despite its efficacy, TNF inhibitors therapy has been shown to increase the incidence of infections in general and of serious infections (such as tuberculosis) in particular.[14,15]

In 2016, an estimated 10.4 million incident cases of tuberculosis (TB) will have occurred worldwide.[16] Brazil ranks 20th among the 30 countries with the highest TB burden in the world, and accounts for approximately one-third of all incident cases of TB in the Americas. [17] In 2017, TB incidence rate of 33.5 cases per 100,000 population were reported in Brazil. Rates in the state of Rio Grande do Sul (RS) appear to be higher than the national average, with an incidence rate of 39.5 cases per 100,000 population; this makes RS one of the four states with the highest incidence of new-onset TB in Brazil.[18]

The risk of developing TB is higher in individuals with RA when compared to the general population.[19] This risk is fourfold higher in RA patients on TNF inhibitors therapy when compared to that of anti-TNF-naive RA patients. [15,20–22] In areas with a higher incidence of TB, such as in Asian countries, a nearly 26-fold greater risk of TB was found in those exposed to TNF inhibitors agents.[23] In patients given infliximab, risk could be up to 30 times greater than in the general population.[24,25]

Considering this increased risk, screening and treatment of latent tuberculosis infection (LTBI) has been recommended prior to initiation of TNF inhibitors therapy.[2,26–29] Screening for LTBI has been shown to reduce the risk of TB reactivation. According to the Spanish Society of Rheumatology registry of patients on immunobiologicals, BIOBADASER, this practice has managed to reduce the number of incident cases of TB in patients starting TNF inhibitors TNF therapy by 78% between 2002 and 2006, when it was implemented.[29,30]

A previous study carried out in Brazil, based on records from the Brazilian Registry of Biologic Therapy Monitoring (BiobadaBrasil) maintained by the Brazilian Society of Rheumatology, found a TB incidence in RA patients with TNF inhibitors exposure of 2.8 cases per 1,000 exposed.[31] Given the higher-than-average incidence of TB in southern Brazil, it is believed that the rate of new TB cases in patients with rheumatic diseases exposed to TNF inhibitors

therapy in Rio Grande do Sul may also be significantly higher than that of the general population.[18] Within this context, the primary objective of this study is to estimate the incidence of tuberculosis in patients receiving TNF inhibitors therapy for rheumatic diseases. As secondary objectives, we sought to evaluate mortality and the influence of screening for latent tuberculosis infection on clinical outcomes in this population.

## Methods

### Study design and patients

This population-based retrospective cohort study included all Public Health System patients from the state of Rio Grande do Sul who were prescribed and dispensed TNF inhibitors therapy for rheumatic diseases from 2006 to 2016. Inclusion was based on the records of the statewide Exceptional Circumstance Drug Dispensing Program, considering those ICD-10 codes covered by the Brazilian Clinical Protocols and Therapeutic Guidelines for rheumatic diseases. Because TB is a notifiable disease in Brazil, definition of TB cases was based on TB reporting data, obtained from the Notifiable Diseases Information System. This system includes all patients with a confirmed diagnosis of tuberculosis, as well as the clinical presentation, treatment regimen and final follow-up outcome. Positive cases being defined by patients with clinical diagnosis associated with bacteriological diagnosis.

Mortality data were obtained from the Mortality Information System, provided by the state centralizing agency for all death certificates in this state. To unify information from these different databases, the linkage technique was used, whereby standardized information is used to find the same individual across several data sources. The variables used for linkage were patient's name, mother's name, and date of birth.

Screening for latent tuberculosis prior to the initiation of TNF inhibitors is recommended for all patients who are candidates for therapy and drug release is required. In subgroup of patients being followed at the outpatient rheumatology clinic of Hospital de Clínicas de Porto Alegre (HCPA) underwent a tuberculin skin test prior to initiation of TNF inhibitors therapy; if LTBI was detected, positive tuberculin skin test or suggestive chest X-ray findings without history of previous tuberculosis treatment, isoniazid treatment was indicated. This information was obtained from a review of patients' medical records. In addition, patients with and without a history of treatment for LTBI were evaluated for potential TB infection.

### Statistical analyses

Data were analyzed in SPSS version 18.03.

No sample calculation was performed since all samples were scheduled to be included in the study.

Descriptive analyses of the incidence rate of tuberculosis, calculated per 1000 exposed patient-years and per 100,000 exposed patients, were carried out. Person- years were calculated from the first day of TNF inhibitor therapy to the date of tuberculosis infection, or death or discontinuation of TNF inhibitor use.

For incidence density analysis, we used the time of use related to the first exposure for calculation of incidence related to the first exposure, also the total time of use of any TNF inhibitor during follow-up to define cumulative incidence density over the course of the study.

In addition, descriptive analyses of all data related to age, sex, underlying disease, duration of anti-TNF exposure, and number of anti-TNF agents used were performed. Among TB cases, we evaluated demographic characteristics, underlying rheumatic disease, TNF inhibitors agent used, and time elapsed from initiation of TNF inhibitors therapy to onset of TB.

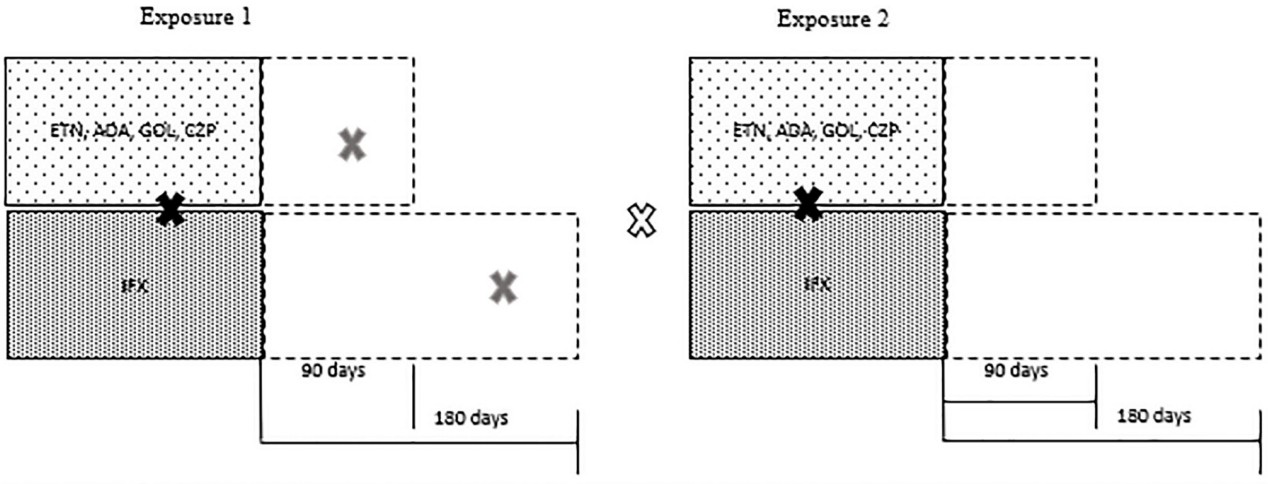

**Fig 1. Model for drug-related tuberculosis cases.** ✖ Cases of TB during TNF inhibitors use. ✖ Cases of TB associated with recente use of TNF inhibitors. ✖ Cases of TB not related to TNF inhibitors use.

Fig 1 illustrates the model used to validate a TB case as related to TNF inhibitors TNF agent exposure. A case of TB was considered related to anti-TNF therapy if it occurred up to 90 days after ETN, ADA, GOL, or CZP were last dispensed to the patient, and up to 180 days after IFX was last dispensed. The 90-day period was defined based on a previous study to evaluate the incidence of tuberculosis, and time on drug influence was still considered.[32] Specifically, considering infliximab this time was extended to 180 days in view of some evidence pointing to the presence of drug circulating up to 28 weeks post exposure.[33]

To compare the incidence of TB among different drugs, incidence was evaluated as percentage, with Fisher's exact test for statistical significance. P-values <0.05 were considered statistically significant.

Disease-free survival time was analyzed in all patients receiving TNF inhibitors therapy by the Kaplan–Meier method, with head-to-head comparison of drugs using the Mantel-Cox log-rank test for the event of interest.

Patient survival was also evaluated using Kaplan–Meier analysis, which included all patients in the cohort. The last evaluation of mortality records was defined as the time point for censoring. The log-rank test was used to compare all-cause mortality versus deaths due to TB.

In the subgroup of patients who underwent screening for LTBI, categorical comparison was performed using Pearson's chi-square test or Fisher's exact test. Again, p-values <0.05 were considered significant for all analyses.

This study was approved by the Research Ethics Committees of Hospital de Clínicas de Porto Alegre and the School of Public Health/Rio Grande do Sul State Department of Health with opinion number 66184417.8.3001.5312.

## Results

A total of 5853 patients and 6902 courses of TNF inhibitors therapy were included, from 2006 to 2016. The sample was predominantly female (68.7%), with a mean age of 49.1 years (Table 1). All five TNF inhibitors agents approved for use in Brazil by the Public Health System were covered: adalimumab accounted for 2980 courses of treatment (43.8%), etanercept for

**Table 1. Demographic and clinical features in patients in patients exposed to TNF inhibitors.**

| Patients' features | RA (n = 3653) | AS (n = 1150) | PsA (n = 872) | JIA (n = 124) | Other spondyloarthritis (n = 54) | Total (n = 5853) | p* |
|---|---|---|---|---|---|---|---|
| Female | 3004 (82.2)* | 467 (40.6) | 446 (51.1) | 76 (61.2) | 29 (53.7) | 4022 (68.7) | <0,001 |
| Age (years)—mean (SD) | 52.7 (12.8)[d] | 42.8 (11.9)[b] | 47.7 (12.5)[c] | 13.1 (7.51)[a] | 44.0 (13.5)[bc] | 49.1 (14.1) | <0,001 |
| ≤65 years | 3039 (83.2) | 1112 (96.7)* | 798 (91.5)* | 124 (100)* | 51 (94.4) | 5124 (87.6) | |
| >65 years | 614 (16.8)* | 38 (3.3) | 74 (8.5) | 0 | 3 (5.6) | 729 (12.4) | |
| TNF inhibitors agent | 4212 | 1345 | 1030 | 151 | 60 | 6802 | |
| IFX | 430 (10.2)* | 113 (8.4) | 69 (6.6) | 7 (4.6) | 4 (6.6) | 623 (9.1) | 0,001 |
| ETN | 1441 (34.1) | 533 (39.2)* | 427 (41.4)* | 118 (78.1)* | 24 (40) | 2543 (37.3) | <0,001 |
| ADA | 1695 (40.2) | 696 (51.74)* | 529 (51.3)* | 28 (18.5) | 32 (53.3) | 2980 (43.8) | <0,001 |
| GOL | 483 (11.4)* | 3 (0.2) | 4 (0.3) | 2 (1.3) | 0 | 492 (7.2) | <0,001 |
| CZP | 163 (3.8)* | 0 | 1 (0.1) | 0 | 0 | 164 (2.4) | <0,001 |
| Number of patients per TNF inhibitors | | | | | | | 0,006 |
| 1 TNF inhibitors | 3090 (84.6)* | 953 (83) | 713 (81.7) | 92 (74.2) | 48 (89) | 4896 (83.6) | |
| 2 or more TNF inhibitors | 563 (15.4) | 197 (17) | 159 (18.3) | 32 (25.8)* | 6 (11) | 957 (16.4) | |
| Duration of TNF inhibitors therapy—mean years (SD) | 2.55 (2.03)[b] | 2.67 (1.71)[b] | 2.59 (1.85)[b] | 2.87 (2.27)[b] | 1.56(1.55)[a] | 2.58 (1.95) | <0,001 |

Abbreviations: RA, rheumatoid arthritis; AS, ankylosing spondylitis; PsA, psoriatic arthritis; JIA, juvenile idiopathic arthritis; TNF, tumor necrosis factor; IFX, infliximab; ETN, etanercept; ADA, adalimumab; GOL, golimumab; CZP, certolizumab pegol; SD, standard deviation;

* p significant

variables with letters not repeated = p significant

2543 (37.4%), infliximab for 623 (9.1%), golimumab for 492 (7.2%), and certolizumab pegol for 164 (2.4%).

Of the patients included, 3653 (62.4%) had been diagnosed with rheumatoid arthritis, 1150 (19.7%) with ankylosing spondylitis, 872 (14.9%) with psoriatic arthritis, 124 (2.1%) with juvenile idiopathic arthritis, and 54 (0.9%) with other inflammatory spondyloarthropathies.

In this sample of TNF inhibitors users, 4896 (83.6%) used only one TNF inhibitor throughout the study period, while 957 (16.4%) were exposed to two or more drugs of this class. The average duration of follow-up, ie exposure to TNF inhibitors therapy was 2.58 ± 1.95 years.

Of the 5853 patients included, 43 received a diagnosis of TB during follow-up; 28 of these occurred during the first exposure. The characteristics of these patients are described in Table 2. The mean age of these patients was 49.5 ± 14.7 years. No significant relationship was found with age >65 years (p = 0.298). Twenty-four (55.8%) were women, 28 (65.1%) had RA, 8 (18.6%) had AS, 6 (13.9%) had PsA, and 1 (2.3%) had JIA. Regarding anti-TNF agents, 27 (62.8%) cases were associated with ADA, 10 (23.2%) with ETN, 3 (7%) with IFX, 2 (4.6%) with CZP and 1 (2.3%) with GOL.

The overall incidence rate of TB in the study population was 734.7 cases per 100,000 exposed. In patient-years, this incidence corresponds to 2.73 per 1000 patient-years exposed (considering first exposure to an TNF inhibitor agent), while the cumulative incidence of TB in the overall study population was 2.86 per 1000 patient-years. Disaggregated by rheumatic disease, the cumulative incidence was 3 per 1000 patient-years for RA (9305 patient-years exposed), 2.61 per 1000 patient-years for AS (3054 patient-years exposed), 2.66 per 1000 patient-years for PsA (2249 patient-years exposed), and 2.8 per 1000 patient-years for JIA (353 patient-years exposed), with no significant differences across groups.

Fig 2 shows the number of TB cases among the different TNF inhibitors agents in relation to the total number of cases, with 0.7% of cases occurring with ADA, 1.5% with CZP, 0.2%

**Table 2. Demographic and clinical features in patients with tuberculosis.**

| Patients with tuberculosis | RA (n = 28) | AS (n = 8) | PsA (n = 6) | JIA (n = 1) | Total (n = 43) | p |
|---|---|---|---|---|---|---|
| Female | 19 (67.8) | 4 (50) | 1 (16.7) | 0) | 24 (55.8) | 0,080 |
| Age (years)—mean (SD) | 52 (14.1) | 43.5 (11.1) | 51.6 (14.3) | 13.4 | 49.5 (14.7) | 0,298 |
| ≤65 years | 21 (75) | 7 (100) | 5 (83.3) | 1 (100) | 35 (81.4) | |
| >65 years | 7 (25) | 0 | 1 (16.7) | 0 | 8 (18.6) | |
| TNF inhibitors agent | | | | | | |
| IFX | 3 (10.7) | 0 | 0 | 0 | 3 (7) | 0,501 |
| ETN | 6 (21.4) | 2 (25) | 1 (16.7) | 1 (100) | 10 (23.2) | 0,344 |
| ADA | 16 (57.1) | 6 (75) | 5 (83.3) | 0 | 27 (62.8) | 0,344 |
| GOL | 1 (3.6) | 0 | 0 | 0 | 1 (2.3) | 0,771 |
| CZP | 2 (7.1) | 0 | 0 | 0 | 2 (4.6) | 0,771 |
| Patients with prior TNF inhibitors | 11 (37.4) | 0 | 4 (66.7)* | 0 | 15 (34.9) | 0,028 |
| Time to active TB§ (years)—mean (SD) | 0.5 (0.1) | 0.9 (0.7) | 1.3 (1.1) | 1.7 | 0.9 (0.5) | 0,981 |
| Site of TB | | | | | | 0,678 |
| Pulmonary | 17 (60.7) | 5 (62.5) | 3 (50) | 1 (100) | 26 (60.5) | |
| Extrapulmonary | 7 (25) | 3 (37.5) | 3 (50) | 0 | 13 (30.2) | |
| Pulmonary and extrapulmonary | 4 (14.3) | 0 | 0 | 0 | 4 (9.3) | |

Abbreviations: RA, rheumatoid arthritis; AS, ankylosing spondylitis; PsA, psoriatic arthritis; JIA, juvenile idiopathic arthritis; TNF, tumor necrosis factor; IFX, infliximab; ETN, etanercept; ADA, adalimumab; GOL, golimumab; CZP, certolizumab pegol; SD, standard deviation.

§Time from onset of TNF inhibitors to development of tuberculosis

* p significant

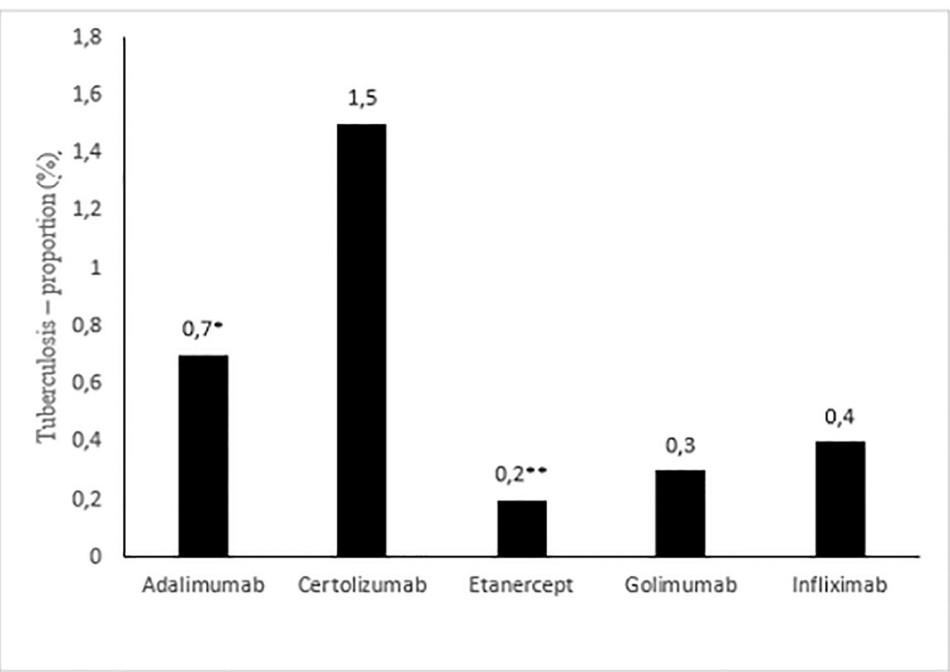

**Fig 2. Tuberculosis—Proportion related to first exposure to TNF inhibitors in percentage.** *Positive association with tuberculosis. **Negative association with tuberculosis. p = 0.043, Fisher's exact test.

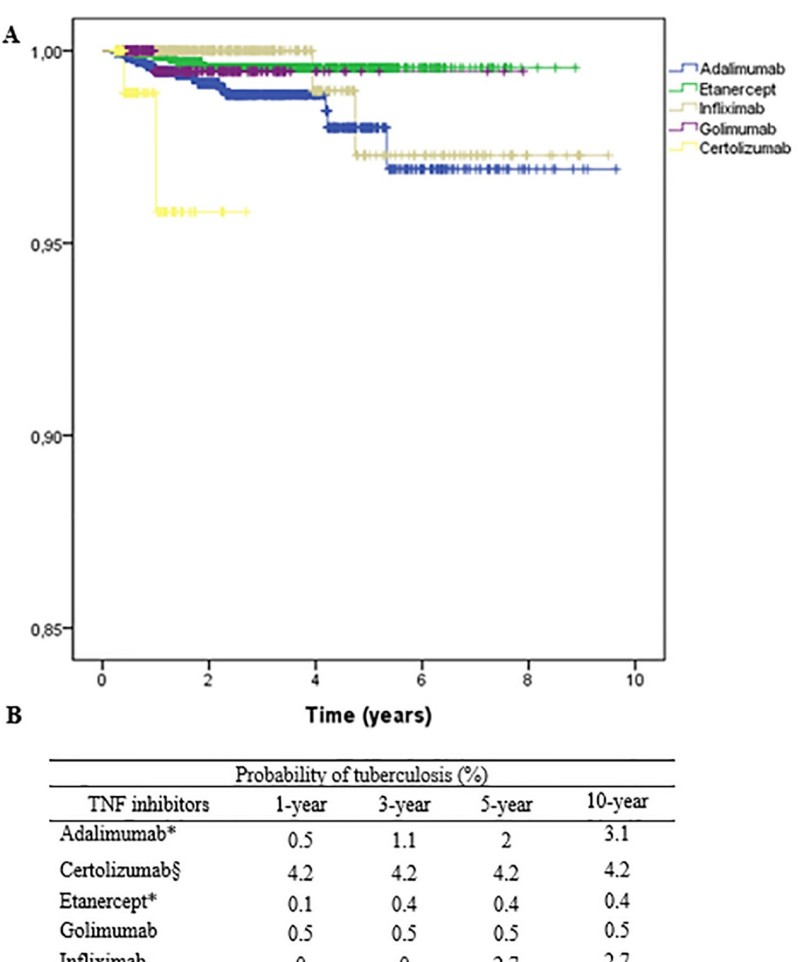

**Fig 3. A:** Survival curves for patients on tumor necrosis factor antagonists as a function of whether they developed tuberculosis during follow-up (log-rank p < 0.05). **B:** Probability of tuberculosis considering disease-free drug use in years (* p< 0.02; § p<0.01 for all TNF by log rank test).

with ETN, 0.3% with GOL, and 0.4% with IFX. The proportion of cases of TB was significantly higher among ADA users than among ETN users (p = 0.043, Fisher's exact test). Fifteen patients had prior exposure to another TNF inhibitor agent. The mean time elapsed from initiation of TNF inhibitors therapy to onset of TB was 0.9 ± 0.5 years overall (i.e., for all anti-TNF agents).

The time tuberculosis-free drug use (Fig 3) differed significantly across different TNF inhibitors agents, with a higher likelihood of developing TB at first exposure with ADA versus ETN (p = 0.01) and a shorter time to onset of TB with CZP therapy than with any other TNF inhibitors agent (p < 0.05).

The variables gender, age, duration of exposure, number of exposures, rheumatic disease and exposure medication were evaluated in the regression model. When performing multivariate Poisson regression analysis, the number of exposures remained associated with the development of TB (RR = 1.17, 95% CI 1.01 to 1.37, p = 0.043) indicating that for each exposure the incidence increased of TB increased by 17%. The first exposure medication also remained after Poisson regression significantly associated with the risk of developing tuberculosis, among medications, ADA presented a 3.11-fold higher risk of TB when compared to ETN (RR = 3.11,

95% CI 1.16 to 8.35, p = 0.024), in addition to this medication, the use of CZP increased the risk of tuberculosis in 7.47-fold compared to ETN (RR = 7.47, 95% CI 1.39–40.0, p = 0.019). The other medications were not significantly different from the risk of ETN (p> 0.5).

The subsample designed to evaluate LTBI treatment enrolled patients from the assisted therapy center of the outpatient rheumatology clinic of Hospital de Clínicas de Porto Alegre. A total of 268 patients in this subgroup were on TNF inhibitors therapy, among these patients, 232 (86%) underwent a tuberculin (PPD) skin test before initiation of TNF inhibitors therapy; 70 (30.2%) tested positive (wheal >5mm). Ninety-three patients underwent repeat tuberculin skin test, of whom 3 ultimately converted to a reactive result. Of the 70 patients with a reactive PPD test, 43 (23.3%) had RA, 20 (50%) had AS, and 7 (24.1%) had PsA; patients with AS were significantly more likely to test positive than patients with RA or PsA (p = 0.004). Seventy-four patients completed treatment for latent TB. There were 5 cases of TB diagnosed during TNF inhibitors therapy. Of these patients, 2 had completed treatment for LTBI, 2 had been nonreactive on PPD skin test, and 1 had not undergone PPD testing (a PPD test had been performed more than 2 years before initiation of TNF inhibitors therapy). The number of cases of tuberculosis was compared between groups with treatment for latent tuberculosis, non-reactive PPD and PPD not known without significant difference between the groups (p = 0.379, p = 0.700 respectively by Fischer's exact test). There was no significant difference in cases of tuberculosis across these different rheumatic diseases.

Regarding all-cause mortality, there were 250 deaths in the overall sample (4.3%). Among patients with TB, 2 (both with RA) had a TB-related death. As shown in Fig 4, the mean 10-year overall survival rate of these patients by the Kaplan–Meier method was 95.7%, with no statistically significant difference on the log rank test (p = 0.2) among all-cause deaths and deaths from tuberculosis.

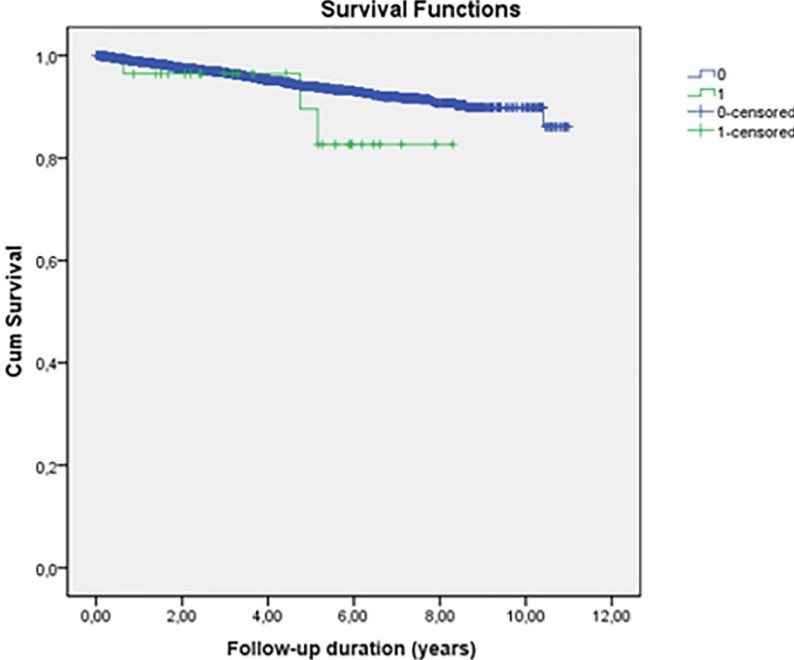

**Fig 4. Survival in relation patients with TNF inhibitors use with deaths due to general causes and deaths due to tuberculosis.** (p = 0,20 by log rank test). 0 = deaths from all causes. 1 = deaths due to tuberculosis.

## Discussion

Brazil is a country with high TB incidence, particularly so in Rio Grande do Sul.[18] Our study found a TB incidence at first anti-TNF exposure of 2.73 per 1000 patient-years exposed. This rate is similar to that found in a previous Brazilian study of RA patients (2.86 per 1000 patient-years)[31] and in studies conducted in the first years of TNF inhibitors prescribing in countries such as Italy and Canada, which reported incidence rates of 2.46 per 1000 and 2.57 per 1000 patient-years exposed, respectively. [14,34] These countries had a much lower incidence of TB in the general population than Brazil. However, as the studies were conducted at the very dawn of TNF inhibitors therapy for the treatment of rheumatic diseases, protocols for LTBI screening and treatment were not yet well established, and the risk of reactivation of granulomatous disease was consequently higher, with a greater number of cases in their populations. The implementation of LTBI therapy revealed a 7-fold higher likelihood of TB reactivation in those exposed to TNF inhibitors therapy who did not follow guidelines for latent TB treatment, and a 78% reduction in the rate of active TB cases among those who did [29,35].

In this sense, our subgroup analysis of patients from the HCPA outpatient rheumatology clinic evaluated LTBI screening and treatment prior to the start of TNF inhibitors therapy. The tuberculin test is recommended in our country for latent tuberculosis evaluation, although it is known that Interferon Gamma Release Assay has better reproducibility and specificity than the tuberculin test, but not yet defined protocol of use in areas with higher incidence of tuberculosis in the general population. [36,37]. A tuberculin test was performed in 86% of patients prior to initiation of immunobiologics, and approximately 30% tested positive, a rate similar to those described elsewhere in the literature.[29,38,39] Comparison across different rheumatic diseases revealed a significantly higher rate of PPD positivity among individuals with AS compared to those with RA or PsA; this finding is consistent with the well-known lower responsiveness of RA patients to tuberculin.[40] However, despite this greater anergy to the Mantoux reaction and consequent risk of false-negative results, this group of patients with RA did not have a higher risk of active TB. Ultimately, there was no significant difference in the rate of incident TB infection among patients with different rheumatic diseases.

As in previous investigations, our study showed a high incidence of TB in patients with rheumatic diseases exposed to TNF inhibitors therapy: 735.29 per 100,000 exposed, approximately 18 times higher than that of the general population of the state of Rio Grande do Sul (39.5 per 100,000).[24]

Considering the different TNF inhibitors agents in use, a higher incidence of TB at first exposure was associated with ADA and CZP therapy, while the lowest incidence was found among patients on ETN. It bears stressing that the only statistically significant difference found on head-to-head comparison was between ADA and ETN, with a positive association of TB cases with ADA therapy and a negative association between ETN therapy and onset of TB, with a risk about 3 times greater of TB development when compared to ETN. This differential risk of TB development in relation to ETN therapy has been reported in previous studies.[7] The lower risk of TB in ETN users has been associated with its mechanism of action. ETN tends to bind to TNF in a less stable manner, which could be associated with less structural damage to granuloma, especially when compared to ADA and IFX.[41–43] We found a trend toward higher incidence of TB among patients on CZP. However, the difference was not statistically significant, probably due to the small number of patients on this drug; however, an interesting—and statistically significant—finding regarding CZP therapy was that it was associated with the highest likelihood of TB development in the first year of exposure compared to the other TNF inhibitors agents. In addition to presenting about 7 times greater risk of developing TB when compared to ETN. Some studies have noted a trend toward frequent TB cases

with CZP therapy, but real-life studies of this drug are scarce due to its more recent introduction in relation to other TNF inhibitors agents. Nevertheless, data from clinical trials also support an association with a high number of TB cases.[44] To date, studies with BiobadaBrasil registry data have not shown any incident cases of TB in users of GOL and CZP, unlike in our cohort.[31,45]

In addition to the risk associated with the type of TNF inhibitor, another feature that can be assessed due to the design of our study was the increased risk of developing tuberculosis related to the number of exposures to TNF inhibitors, with a 17% increase in risk each time. although exposure time was not at higher risk as in previous studies [46].

Considering the different rheumatologic diseases evaluated, in our study we found no statistically significant difference between the incidence of tuberculosis in the different rheumatologic pathologies that were under study. Despite the increased risk of developing tuberculosis in rheumatoid arthritis patients compared to the general population, when comparing the risk of tuberculosis with the use of TNF inhibitors among inflammatory arthropathies, our study and previous studies found no significant difference. [24,47]

Regarding the mean time to TB development, active granulomatous disease was detected already in the first year of exposure to TNF inhibitors therapy (mean time elapsed, 0.9 ± 0.5 years). This is corroborated by several studies which have suggested a higher risk of TB development in the first months of TNF inhibitors therapy, a phenomenon attributable to reactivation of latent disease.[46,48–50] In addition, considering the disease-free survival time, there was a significantly higher likelihood of developing TB over the years in patients exposed to ADA than in those exposed to ETN, as well as a significantly higher likelihood of TB onset in the first year of exposure among CZP users in relation to all other TNF inhibitors agents.

Analysis of TB sites revealed a predominance of pulmonary tuberculosis (n = 26, 60.4%), as in previous studies.[34,51] Extrapulmonary TB was found in 13 cases (30.2%), a rate lower than that reported in previous studies of national registry data.[31,50]

In this cohort of TNF inhibitors users, the overall survival rate was 95.7%. All-cause mortality was thus 4.3% over the study period. Among cases of incident TB, there were 2 deaths attributable to TB infection (4.6%). These findings are similar to previous reports of mortality data among TB patients receiving TNF inhibitors therapy. TB-attributable mortality rates are consistently lower than in the general population, probably due to the presence of HIV-coinfected patients in the general population.[52,53]

The present study is the first large investigation to evaluate the incidence of TB in patients receiving TNF inhibitors therapy for rheumatic diseases in the state of Rio Grande do Sul, and one of the first in Brazil to evaluate the incidence of TB in such a large population (n = 5853) with different rheumatic conditions over such a long follow-up period, using real-life, population-based data.

Some limitations of this study are inherent to retrospective designs. As our data were retrieved from existing records, no information could be obtained on LTBI screening and treatment prior to initiation of TNF inhibitors therapy, except in our subgroup analysis. In addition, exposure to TNF inhibitors agents was assessed solely by the proxy indicators of drug dispensing and discontinuation. We did not analyze data on concomitant use of other disease-modifying antirheumatic drugs, for instance, nor did we assess whether TNF inhibitors therapy was used in combination with glucocorticoids, which could constitute an additional risk factor for development of TB in this population. In addition to these limitations in our study we were unable to evaluate possible host-related risk factors for TB such as drug or alcohol use, associated comorbidities, and socioeconomic conditions. We also had fewer prescriptions of infliximab and golimumab, and their negative association with tuberculosis should be interpreted with caution. Nevertheless, this was the largest retrospective cohort

study of patients receiving TNF inhibitors therapy for rheumatic diseases conducted to date in Brazil. Its retrospective design notwithstanding, the fact that TB and mortality records were obtained from SINAN and from the SUS mortality information system respectively ensures that these data were reliable, because TB is a notifiable disease in Brazil and the associated mortality data were fed into the mortality information system from official death certificates. Besides that, our study included only patients using the public health system, not including patients with TNF inhibitors prescription through health insurance or private acquisition, and may be a selection bias, however, a Brazilian study showed in the sample that the portion of patients with health insurance represented only about 15% of immunobiological prescriptions.[45]

## Conclusions

The use of TNF inhibitors therapy by patients with systemic rheumatic diseases significantly increases the number of incident cases of tuberculosis in this population, with an 18-time increased TB incidence in these patients when compared to the general population. Considering the different TNF inhibitors agents used by our sample, our study demonstrated that a greater number of cases of TB were associated with ADA than with ETN, Finally, we found that newer drugs such as CZP, which have had a much shorter postmarketing surveillance period for assessment of their potential for infectious adverse events, appear to be associated with a higher frequency of incident TB. This highlights the need for proper screening and treatment of latent tuberculosis, as well as the surveillance of new contacts with individuals with tuberculosis, annual screening among people with initial negative screening as recent recommendations, and discussion of the use of interferon gamma release quantification tests more broadly in this population.

## Acknowledgments

We would like to thank all the individuals who participated in any way that this study has achieved its outcome. Special thanks to the teams of the state health department who worked to make all the data used in the research feasible.

## Author Contributions

**Conceptualization:** Natália Sarzi Sartori, Rafael Mendonça da Silva Chakr.

**Data curation:** Natália Sarzi Sartori.

**Formal analysis:** Jeruza Lavanholi Neyeloff, Rafael Mendonça da Silva Chakr.

**Investigation:** Natália Sarzi Sartori, Afonso Papke, Rafael Mendonça da Silva Chakr.

**Methodology:** Natália Sarzi Sartori, Paulo Picon, Afonso Papke, Jeruza Lavanholi Neyeloff, Rafael Mendonça da Silva Chakr.

**Project administration:** Rafael Mendonça da Silva Chakr.

**Resources:** Afonso Papke.

**Software:** Jeruza Lavanholi Neyeloff.

**Supervision:** Natália Sarzi Sartori, Rafael Mendonça da Silva Chakr.

**Validation:** Natália Sarzi Sartori.

**Visualization:** Natália Sarzi Sartori.

**Writing – original draft:** Natália Sarzi Sartori, Afonso Papke, Rafael Mendonça da Silva Chakr.

**Writing – review & editing:** Paulo Picon, Rafael Mendonça da Silva Chakr.

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
