## [Decision Letter · Decision Letter 0]

2 Sep 2019

PONE-D-19-21256

A POPULATION-BASED STUDY OF TUBERCULOSIS INCIDENCE AMONG RHEUMATIC DISEASE PATIENTS UNDER ANTI-TNF TREATMENT

PLOS ONE

Dear Mrs Sartori,

Thank you for submitting your manuscript to PLOS ONE. After careful consideration, we feel that it has merit but does not fully meet PLOS ONE’s publication criteria as it currently stands. Therefore, we invite you to submit a revised version of the manuscript that addresses the points raised during the review process.

Please address all comments of the Reviewers and all of the statistical analyses need to  be reviewed by a statistician 

We would appreciate receiving your revised manuscript by Oct 17 2019 11:59PM. To enhance the reproducibility of your results, we recommend that if applicable you deposit your laboratory protocols in protocols.io, where a protocol can be assigned its own identifier (DOI) such that it can be cited independently in the future. For instructions see: http://journals.plos.org/plosone/s/submission-guidelines#loc-laboratory-protocols

We look forward to receiving your revised manuscript.

Kind regards,

Mahmoud Abu-Shakra, MD

Academic Editor

PLOS ONE

Journal Requirements:

Reviewers' comments:

Reviewer's Responses to Questions

**Comments to the Author**

1. Is the manuscript technically sound, and do the data support the conclusions?

Reviewer #1: Yes

Reviewer #2: Yes

2. Has the statistical analysis been performed appropriately and rigorously? 

Reviewer #1: Yes

Reviewer #2: Yes

3. Have the authors made all data underlying the findings in their manuscript fully available?

Reviewer #1: Yes

Reviewer #2: Yes

4. Is the manuscript presented in an intelligible fashion and written in standard English?

Reviewer #1: Yes

Reviewer #2: Yes

5. Review Comments to the Author

Reviewer #1: General comment.

The paper confirm the increased risk of TB reactivation associated with anti-TNFs in and endemic TB Country. Apart from some English language imprecisions that should be corrected, the report is exhaustive and provides useful data for clinicians.

Specific comments.

Page 3. Line 2. References 1 and 2 should be written in superscript.

Page 4. First sentence. The Authors should correct this phrase by adding that RA itself increases the risk of TB (add the referencei.e. Yeh JJ, et al. PLoS One. 2014 Oct 22;9(10):e110922.

Page 5. Line 3 and followings, The AA. should better address the notification system in Brazil. In some Countries also the suspected cases of TB should be notified. Is the same in Brazil, or are only definite TB diagnoses included? This issue is not negligible in terms of TB incidence calculation. Does the system include information from the patients' clinical chart?

Page 5. Last paragraph. Apart from the cohort observed at the Hospital of Porto Alegre, did the patients undergo LTBI screening? The AA. should indicate this.

Page 5. Last paragrafh, The AA. should indicate if LTBI diagnosis was done by TST only, or, at least in a percentage of patients, by Quantiferon, combined or not with TST.

Page 10. Line 10. The AA. should insert a sentence on the better accuracy of Quantiferon TB Plus test for LTBI screening. (see. Baddley JW, et al. Clin Microbiol Infect. 2018 Jun;24 Suppl 2:S10-S20.

Page 12. Discussion section. Last paragraph. Regarding the study limits, the AA, should add the absence of information on the host-related additional risk factors for TB, including drug or alcohol abuse, comorbidities, socio-economic conditons, etc.

Page 13. Conclusions. The text is redundant and with some repetitions. I suggest to shorten this section by underlining the main results and emphasizing the need of accurate LTBI screening (adding Quantiferon-TB Plus test).

Reviewer #2: This was a cohort study of tuberculosis in rheumatic disease patients who have been treated with anti-TNF. It is useful to get estimates from other parts of the world, especially those with a higher incidence in the underlying population, to better understand what the risk is of this adverse outcome. The linkage of data across registers in Brazil was an excellent way to answer this study question. There are some basic methodological and epidemiological pieces of the study that were not described that need clarification.

Major comments

- The TB cases were based on reporting data. Did these reports depend on TB being culture-verified? Could any TB be reported to the system or did it have to be reported through a lab that found a positive test?

- The study design and patients does not describe person-time (start of follow-up, end of follow-up). Given that there are incidence rates and survival estimates in the paper, how person-time was calculated needs to be included.

- Is this a new-user design? Are the exposures to TNFi all instances of the 1st use of TNFi? Or 1st use of the particular TNFi? This should be made more clear in the methods. In the results it states that the number of exposures was used as a variable in the model, so this indicates that people had previous exposures – of the same TNFi or another TNFi?

- It is odd that incidence is “evaluated as a percentage” when incidence is a rate – the number of cases, per person per year. There is an aspect of time that is missing. It is hard to compare these results to those from other studies when percentages are given.

- A summary of the follow-up time should be included in the results (see STROBE guidelines, if needed)

- Are the survival curves adjusted for anything?

- A table 1 in a cohort study usually has the exposure as the header of the table so that the reader can see how characteristics of the study differ by exposure. This is necessary for the reader to understand other factors, such as age and sex, which are potentially related to starting on a TNFi.

- To have p values for every line of every table and every line in every graph is a lot of tests and does not seem necessary (for example, we can see clearly that the age is different across rheumatic diseases in Table 1, it’s not necessary to do a statistical test for it).

- In Figure 2, it says there was a positive association, compared to what? These are unadjusted for age and sex, which is a major limitation. The reader is interested in rates (unadjusted are ok, sex- and age-standardized are even better) and then the adjusted rate ratios are helpful for comparing rates. Statistically comparing unadjusted rates is not very useful for inference.

- There wasn’t much discussion about potential differences in TB risk in the different rheumatic diseases. This could be a limitation if all diseases are put together in this study.

Minor comments

- In the abstract results there is a typo, it should be “Forty-three cases”

- In the discussion, one usually starts with a first sentence that summarizes the results. It would be a better start to the discussion if the first sentence (describing a recent study) is moved elsewhere and the results of the present study are summarized.

6. PLOS authors have the option to publish the peer review history of their article (what does this mean?). If published, this will include your full peer review and any attached files.

Reviewer #1: No

Reviewer #2: No

---

## [Author Response · Author response to Decision Letter 0]

17 Oct 2019

Dear Publishers,

We respectfully welcome and appreciate the valuable contributions of the reviewers, with their suggestions and correction requests that certainly qualify and give greater consistency to our study.

The following describes the treatment given to each of the changes proposed by the reviewers for resubmission of the manuscript PONE-D-19-21256 (A POPULATION-BASED STUDY OF TUBERCULOSIS INCIDENCE AMONG RHEUMATIC DISEASE PATIENTS UNDER ANTI-TNF TREATMENT).

Reviewer 1:

Introduction:

#Page 3. Line 2. References 1 and 2 should be written in superscript.

Thank you for your observation, the correction in the body of the text (pages 3, lines 1 and 2).

#Page 4. First sentence. The Authors should correct this phrase by adding that RA itself increases the risk of TB (add the referencei.e. Yeh JJ, et al. PLoS One. 2014 Oct

We thank and adjust information about the risks of tuberculosis in this patient population, adding reference suggested by relevance. (page 4, line 1 and 2).

Methods:

#Page 5. Line 3 and followings, The AA. should better address the notification system in Brazil. In some Countries also the suspected cases of TB should be notified. Is the same in Brazil, or are only definite TB diagnoses included? This issue is not negligible in terms of TB incidence calculation. Does the system include information from the patients' clinical chart?

This information has been added to the methods section to clarify the definition of tuberculosis cases in the Brazilian reporting system.

“This system includes all patients with a confirmed diagnosis of tuberculosis, as well as the clinical presentation, treatment regimen and final follow-up outcome.” (Page 5, line 8)

#Page 5. Last paragraph. Apart from the cohort observed at the Hospital of Porto Alegre, did the patients undergo LTBI screening? The AA. should indicate this.

 The request has been fulfilled and included in an attempt to provide clarity.

“Screening for latent tuberculosis prior to the initiation of TNF inhibitors is recommended for all patients who are candidates for therapy and drug release is required.” (page 5 – last paragraph)

#Page 5. Last paragrafh, The AA. should indicate if LTBI diagnosis was done by TST only, or, at least in a percentage of patients, by Quantiferon, combined or not with TST. 

Thank you for your observation, the correction in the body of the text.

“positive tuberculin skin test or suggestive chest X-ray findings without history of previous tuberculosis treatment” (page 5 – last paragraph).

Discussion:

#Page 10. Line 10. The AA. should insert a sentence on the better accuracy of Quantiferon TB Plus test for LTBI screening. (see. Baddley JW, et al. Clin Microbiol Infect. 2018 Jun;24 Suppl 2:S10-S20.

 Relevant information about the use of another method for identifying latent tuberculosis was added to the discussion.

“The tuberculin test is recommended in our country for latent tuberculosis evaluation, although it is known that Interferon Gamma Release Assay has better reproducibility and specificity than the tuberculin test, but not yet defined protocol of use in areas with higher incidence of tuberculosis in the general population” (page 10 – last paragraph).

#Page 12. Discussion section. Last paragraph. Regarding the study limits, the AA, should add the absence of information on the host-related additional risk factors for TB, including drug or alcohol abuse, comorbidities, socio-economic conditons, etc.

Information was mentioned in the discussion as limitations of our study.

“In addition to these limitations in our study we were unable to evaluate possible host-related risk factors for TB such as drug or alcohol use, associated comorbidities, and socioeconomic conditions” (page 12 – last paragraph).

Conclusion:

#Page 13. Conclusions. The text is redundant and with some repetitions. I suggest to shorten this section by underlining the main results and emphasizing the need of accurate LTBI screening (adding Quantiferon-TB Plus test).

The findings have been rewritten to meet the proposed objectives. (page 14 - last paragraph).

Reviewer 2:

Major comments

Methods:

# The TB cases were based on reporting data. Did these reports depend on TB being culture-verified? Could any TB be reported to the system or did it have to be reported through a lab that found a positive test?

This information has been added to the methods section to clarify the definition of tuberculosis cases in the Brazilian reporting system.

“Positive cases being defined by patients with clinical diagnosis associated with bacteriological diagnosis.” (page5, line 10)

# The study design and patients does not describe person-time (start of follow-up, end of follow-up). Given that there are incidence rates and survival estimates in the paper, how person-time was calculated needs to be included.

This information was added to the methods in order to provide more clarity in the definitions used.

“Person- years were calculated from the first day of TNF inhibitor therapy to the date of tuberculosis infection, or death or discontinuation of TNF inhibitor use” (page 6, line 3)

# Is this a new-user design? Are the exposures to TNFi all instances of the 1st use of TNFi? Or 1st use of the particular TNFi? This should be made more clear in the methods. In the results it states that the number of exposures was used as a variable in the model, so this indicates that people had previous exposures – of the same TNFi or another TNFi?

 This information was added to the methods in order to provide more clarity in the definitions used. For TNF inhibitor exposure, this is a new-user design. 

“For incidence density analysis, we used the time of use related to the first exposure for calculation of incidence related to the first exposure, also the total time of use of any TNF inhibitor during follow-up to define cumulative incidence density over the course of the study.” (page 6, second paragraph)

Results:

# It is odd that incidence is “evaluated as a percentage” when incidence is a rate – the number of cases, per person per year. There is an aspect of time that is missing. It is hard to compare these results to those from other studies when percentages are given.

The incidence density is presented in person time for better comparison with studies as mentioned (page 8 - second paragraph).

As noted, we corrected information referring to figure 2 in the body of the text and in the figure itself, in which we had no intention of presenting incidence rates and density, but only presentation in the number of cases in proportion.

“Figure 2 shows the number of TB cases among the different TNF inhibitors agents in relation to the total number of cases, with 0.7% of cases occurring with ADA, 1.5% with CZP, 0.2% with ETN, 0.3% with GOL, and 0.4% with IFX. The proportion of cases of TB was significantly higher among ADA users than among ETN users (p = 0.043, Fisher’s exact test)” (page 8 – third paragraph).

#A summary of the follow-up time should be included in the results (see STROBE guidelines, if needed)

Adjustment was performed in the body of the text as suggested and oriented guidelines STROBE.

“The average duration of follow-up, ie exposure to TNF inhibitors therapy was 2.58 ± 1.95 years”. (page 7 – sixth paragraph).

# Are the survival curves adjusted for anything?

 As described in the body text adjustment was performed.

“The variables gender, age, duration of exposure, number of exposures, rheumatic disease and exposure medication were evaluated in the regression model”. (page 8 – last paragraph).

# A table 1 in a cohort study usually has the exposure as the header of the table so that the reader can see how characteristics of the study differ by exposure. This is necessary for the reader to understand other factors, such as age and sex, which are potentially related to starting on a TNFi.

"Thank you for your relevant comment on table 1. In the submitted version we tried to follow STROBE´s recommendation to give characteristics of study participants (eg demographic, clinical, social) and information on exposures and potential confounders, but we are open to modify it, if necessary. In this population-based study, our primary goal was to estimate the incidence of tuberculosis in patients receiving TNF inhibitors (TNFi) therapy for rheumatic diseases. Considering that all participants were exposed to TNFi, we thought it would be more informative to report on how each disease subgroup could differ, as treatment protocols are unique and TNFi exposure could vary significantly among rheumatic diseases. As the number of TNFi exposures is expected to be a confounding factor, we included this variable in regression model and, ultimately, found it as an independent predictor of TB development, as reported in the results section. Nonetheless, if this is not the best way to present our participants, we would be pleased to change it accordingly." 

# To have p values for every line of every table and every line in every graph is a lot of tests and does not seem necessary (for example, we can see clearly that the age is different across rheumatic diseases in Table 1, it’s not necessary to do a statistical test for it).

Thanks for the comment, we are grateful for the observation, however, we understand that keeping the value of p shows the differences more clearly. Despite our understanding, if not necessary, we agree to their withdrawal.

# In Figure 2, it says there was a positive association, compared to what? These are unadjusted for age and sex, which is a major limitation. The reader is interested in rates (unadjusted are ok, sex- and age-standardized are even better) and then the adjusted rate ratios are helpful for comparing rates. Statistically comparing unadjusted rates is not very useful for inference.

A) The comparison was made between users of adalimumab and etanercept.

“The proportion of cases of TB was significantly higher among ADA users than among ETN users (p = 0.043, Fisher’s exact test)” (page 8 – third paragraph).

B) We did not perform adjustment in this case because it is not a rate but only the demonstration of the number of tuberculosis cases by each TNF inhibitor. number of exposures, exposure time, type of medication and rheumatologic disease taken to the regression model.

#There wasn’t much discussion about potential differences in TB risk in the different rheumatic diseases. This could be a limitation if all diseases are put together in this study.

As suggested considering the design of our study we included assessment of the incidence density of tuberculosis due to rheumatologic disease separately.

“Disaggregated by rheumatic disease, the cumulative incidence was 3 per 1000 patient-years for RA (9305 patient-years exposed), 2.61 per 1000 patient-years for AS (3054 patient-years exposed), 2.66 per 1000 patient-years for PsA (2249 patient-years exposed), and 2.8 per 1000 patient-years for JIA (353 patient-years exposed), with no significant differences across groups.” (page 8 – second paragraph).

Minor comments 

# In the abstract results there is a typo, it should be “Forty-three cases”

Bug pointed out has been fixed. (abstract - page 2)

# In the discussion, one usually starts with a first sentence that summarizes the results. It would be a better start to the discussion if the first sentence (describing a recent study) is moved elsewhere and the results of the present study are summarized.

The initial discussion session was rewritten and adjusted as suggested for a better presentation of our study findings.

“Brazil is a country with high TB incidence, particularly so in Rio Grande do Sul.18 Our study found a TB incidence at first anti-TNF exposure of 2.73 per 1000 patient-years exposed. This rate is similar to that found in a previous Brazilian study of RA patients (2.86 per 1000 patient-years)31 and in studies conducted in the first years of TNF inhibitors prescribing in countries such as Italy and Canada, which reported incidence rates of 2.46 per 1000 and 2.57 per 1000 patient-years exposed, respectively.” (page 10 – second paragraph)

We are at your disposal for any necessary clarifications.

Sincerely,

The authors

---

## [Decision Letter · Decision Letter 1]

28 Oct 2019

A POPULATION-BASED STUDY OF TUBERCULOSIS INCIDENCE AMONG RHEUMATIC DISEASE PATIENTS UNDER ANTI-TNF TREATMENT

PONE-D-19-21256R1

Dear Dr. Sartori,

We are pleased to inform you that your manuscript has been judged scientifically suitable for publication and will be formally accepted for publication once it complies with all outstanding technical requirements.

With kind regards,

Mahmoud Abu-Shakra, MD

Academic Editor

PLOS ONE

Additional Editor Comments (optional):

Reviewers' comments:

Reviewer's Responses to Questions

**Comments to the Author**

1. If the authors have adequately addressed your comments raised in a previous round of review and you feel that this manuscript is now acceptable for publication, you may indicate that here to bypass the “Comments to the Author” section, enter your conflict of interest statement in the “Confidential to Editor” section, and submit your "Accept" recommendation.

Reviewer #2: All comments have been addressed

2. Is the manuscript technically sound, and do the data support the conclusions?

Reviewer #2: Yes

3. Has the statistical analysis been performed appropriately and rigorously? 

Reviewer #2: Yes

4. Have the authors made all data underlying the findings in their manuscript fully available?

Reviewer #2: Yes

5. Is the manuscript presented in an intelligible fashion and written in standard English?

Reviewer #2: Yes

6. Review Comments to the Author

Reviewer #2: (No Response)

7. PLOS authors have the option to publish the peer review history of their article (what does this mean?). If published, this will include your full peer review and any attached files.

Reviewer #2: Yes: Elizabeth Arkema

---

## [Editor Report · Acceptance letter]

12 Nov 2019

PONE-D-19-21256R1 

A population-based study of tuberculosis incidence among rheumatic disease patients under anti-TNF treatment  

Dear Dr. Sartori:

I am pleased to inform you that your manuscript has been deemed suitable for publication in PLOS ONE. Congratulations! Your manuscript is now with our production department. 

With kind regards,

on behalf of

Dr. Mahmoud Abu-Shakra 

Academic Editor

PLOS ONE